# Impact of the Design Industry on Carbon Emissions in the Manufacturing Industry in China: A Case Study of Zhejiang Province

**Bing Xu * and Haoxiang Qu**

School of Design and Architecture, Zhejiang University of Technology, Hangzhou 310023, China; 201906140306@zjut.edu.cn
* Correspondence: xubing@zjut.edu.cn

**Abstract:** Reducing carbon emissions during the manufacturing process is a top priority for realizing the strategic "carbon neutral" target. Currently, there is a great deal of research about carbon production in the middle and later stages of production. However, studies on upstream processes are relatively scarce. Studies concerning the design stages are particularly uncommon, although this phase has a profound effect on carbon emissions during manufacturing. Therefore, it is vital to study this and related fields in depth. In this paper, we take Zhejiang Province, China, as the research object, and study the relationship between the design industry and carbon emissions in the manufacturing industry. Through analysis, we developed a first-level evaluation index for the influence of design on carbon emissions in the production stage. Then, we subdivided the first-level indexes into several second-level indexes using the entropy method. Subsequently, we calculated their weights and comprehensively evaluated the influence of these design factors on carbon emissions in the manufacturing industry using the ridge regression model test. Results from our research reveal that at the design end of the production chain, the expansion of resource scale of each design phase has a significant inhibitory effect on the carbon emissions of the manufacturing stage. Conversely, improvements to the industrial economic benefit index have a significant positive effect on the carbon emissions in the manufacturing industry, while government support and the innovation composite index have little influence. The main conclusions of this study are as follows: To reduce carbon emissions in the manufacturing industry, the scale of the inhibitory effect of the design process should be fully evaluated, while the scale construction and related resource input of the design phase should be emphasized. Furthermore, guidance regarding the social responsibility of design enterprises should be strengthened to further promote the concept of "green design" and reverse the purely market-oriented focus of the industry. Also, to promote a greener design industry, the provision of high-quality green design talents should be fortified. In the future, an appropriate green design evaluation index system should be devised to ensure the stable economic development of the design industry and challenge the current situation where a focus on economic indexes results in increased carbon emissions.

**Keywords:** design industry; manufacturing; carbon emissions; entropy value method; ridge regression; structural equation model





## 1. Introduction

Since the industrial revolution, along with the population surge and industrial production, human emissions of greenhouse gases have increased by a wide range, and the resulting global climate change poses a major challenge to the development of human society today [1]. In the face of the global climate challenge, China eloquently proposed at the 75th Session of the United Nations General Assembly on 22 September 2020 that it would take effective policies and measures at home. China aims to peak its carbon

dioxide emissions by 2030 and become carbon neutral by 2060 on the basis of increasing its Nationally Determined Contribution (NDC) [2].

To achieve the goal of carbon neutrality, the Chinese government has already issued relevant regulations in finance [3,4], law [5,6], transportation [7,8], production and manufacturing [9–12], and other fields, aiming to comprehensively promote reductions in carbon emission across the country.

The characteristics of high energy consumption and high emissions make manufacturing a vital element in achieving carbon neutrality [13,14]. As the country with the largest manufacturing output in the world [15], China's carbon emissions from manufacturing now account for more than half of its total carbon dioxide emissions. Therefore, achieving the goal of carbon neutrality is a genuine problem that the manufacturing industry must face in the future.

Achieving carbon emission reductions in the manufacturing industry and shifting from the traditional industrial development path to a high-quality development path is a top priority [16–18]. To transform and upgrade the manufacturing industry, academics in China have recognized the following paths: service-oriented manufacturing [19,20], capital-intensive manufacturing [21,22], and design innovation [23–25]. In this paper, we will further study the design-driven path in the transformation and upgrading of the manufacturing industry and focus on the impact of the development of the design industry on carbon emission reduction in the manufacturing industry.

At present, there is much research on the carbon emissions of manufacturing processes [26–29], but few studies on the design phase of the production chain. However, the design stage essentially determines the carbon emissions of the whole production process and the final design output and has a tremendous influence on the carbon emissions of the manufacturing industry. However, since the design industry is an emerging sector, data is insubstantial and difficult to accurately capture and intuitively quantify, thus relevant research is limited [30–32]. This results in a lack of effective regulation of the design industry in the process of formulating carbon emission reduction policies for the manufacturing industry. Also, there is a lack of focus on design industry supervision, which creates loopholes in the regulation of carbon emissions in the manufacturing industry. Therefore, it is necessary to study the impact of the design industry on manufacturing carbon emissions.

To ensure research quality, it is crucial to obtain high-quality data samples. Therefore, in this paper, we selected a region with relatively developed manufacturing and design industries for the data sample. The design industry is an emerging high-tech creative industry with uneven regional distribution in China. However, Zhejiang province is a region with a rapidly developing internet industry and an established retail manufacturing industry, which has a complete range of representative design industries that are relatively uniformly distributed. Therefore, in this paper, we used Zhejiang Province as the research sample. Based on design industry trends and carbon emission data in Zhejiang province from 2016 to 2020, we comprehensively analyzed the design end of the manufacturing industry. The purpose of this paper is to reveal the influence of various evaluation indexes on the development of the design industry and their effect on the carbon emissions of the manufacturing industry.

The research areas involved in this paper include the following two elements: carbon emissions of the manufacturing industry and a development evaluation of the design industry.

Concerning carbon emissions during the manufacturing stage, many scholars have analyzed the driving factors of carbon emissions through the Kaya model and LMDI model [33–38], hoping to reveal the core driving factors of manufacturing carbon emissions. However, these methods are generally only applicable to the analysis of macro factors, such as per capita GDP, energy intensity per unit GDP, etc., and they cannot directly establish specific relationships between the carbon emissions of the manufacturing industry and other related industries. Therefore, in this paper, we adopted different research approaches for the study of the design industry. To measure manufacturing carbon emissions, several studies have used the general Intergovernmental Panel on Climate Change (IPCC) detection

algorithm for carbon dioxide emissions to calculate the emissions generated in production processes through the energy consumption of the manufacturing industry. For example, Nie and Zhou et al. [39] calculated the manufacturing carbon emissions of 11 provinces and cities in the Yangtze River Economic Belt in China from 2004 to 2017 by utilizing the IPCC algorithm. Besides, Wang and Ren [40] used this method to determine the carbon emissions of the manufacturing industry in China from 1995 to 2016. Liu [41] and Li [42] also employed the IPCC algorithm to calculate the carbon emissions of the manufacturing industry in Jiangsu Province from 2003 to 2017 and in Shandong Province from 2000 to 2016, respectively. Research has proved that the IPCC algorithm is a convenient and effective approach for evaluating the carbon emissions of the manufacturing industry based on relevant energy consumption data. Therefore, we can apply this method to calculate the carbon emissions of the manufacturing industry within a province.

In the field of design industry development evaluation, scholars have mostly used mathematical models related to management and statistics. Wang [43] formulated a secondary index and tertiary index for the development evaluation of the design industry by extracting relevant policies issued by the Chinese government at the national and regional level from 2007 to 2015, then calculated the weight of each index through principal component analysis. Qin and Wang [44] established relevant indexes for the competitiveness evaluation of the design industry by referring to evaluation indexes of the creative industry in Europe and Hong Kong. Then, they calculated the weight of each index by applying an analytic hierarchy process. Yan [45] determined first-level, second-level, and third-level indicators of design industry evaluation through a survey and comparative analysis of relevant policies, then determined the weight of each indicator using the Delphi method. Based on the diamond theoretical model and combined with relevant literature and policies, Yu [46] devised eight first-level indicators, 12 second-level indicators, and 59 third-level indicators for the evaluation of cultural creative industry, and proposed that the weight of each indicator should be further measured through a questionnaire and analytic hierarchy process. Chen and Wang et al. [47] obtained the relevant factors of design industry evaluation by combining the diamond model with relevant literature. Subsequently, they analyzed the causality by utilizing the structural equation model.

In summary, relevant studies on the measurement of carbon emissions in the manufacturing sector are relatively comprehensive. On the premise of possessing the relevant data, the carbon emissions of the manufacturing sector can be quickly calculated using the IPCC method. However, in the field of design industry development evaluation, there remain issues of subjectivity, especially regarding the weight calculations of indicators. For instance, the Delphi method and the analytic hierarchy process are both highly subjective since they rely on the evaluation of experts. Besides, no research has focused on the impact of design industry development on manufacturing carbon emissions. Therefore, in this article, we first perform a comprehensive literature analysis to formulate an evaluation index for the development of the design industry, then apply the entropy method to calculate the weight of each index. Also, by using the ridge regression model, we avoid the subjectivity of expert evaluation and preliminary judgment to explain how the design process influences manufacturing carbon emissions. Furthermore, we use the structural equation model to calculate the relationship between each indicator and carbon emissions, then propose a development direction for carbon emission reduction, based on design industry development. This provides theoretical reference and a decision-making foundation for relevant departments and fills the research gap in the field of design industry-influenced carbon emissions.

This paper is divided into four parts. After the introduction, Section 2 introduces the materials used in the research and the mathematical model, Section 3 mainly introduces the calculation results, and Section 4 reviews the research results and draws the conclusions.

## 2. Materials and Methods

### 2.1. Entropy Model

Entropy itself is a concept of thermodynamics, but it was introduced into information theory by Shannon in 1948 [48]. In information theory, entropy is a measure of uncertainty, and calculating the entropy value of an index can judge its degree of dispersion. Greater degrees of dispersion mean there is a larger impact on the final comprehensive evaluation, so we can calculate the weight of each index by using the information carried by its entropy. In this paper, we studied the influence of the design industry on manufacturing carbon emissions by constructing an index system on how the design industry influences these emissions. It is necessary to calculate the weight of each index to define the future development direction of the design industry and more effectively support reductions in manufacturing carbon emissions.

According to existing research [49–51], the design industry evaluation system that affects manufacturing production includes four first-level indicators: design re-source scale index, economic benefit index, government support intensity, and innovation composite index. Since the design industry itself affects carbon emissions by influencing manufacturing production, it is acceptable to use these four indicators as first-level evaluation indexes. Combined with data extracted from the Department of Economy and Information Technology of the Zhejiang Provincial Government, we can further subdivide these four indicators into eight secondary indexes: design companies (home), full-time workers engaged in design (people), number of design transactions (a), output value of design achievement transformation (ten thousand yuan), level of provincial support funding (ten thousand yuan), local matching funding limit (ten thousand yuan), number of authorized patents (a), number of major awards obtained domestically and abroad (a). The number of authorized patents includes the total number of patents for appearance, utility models, and invention. Major awards include the international Red Dot Award and the domestic Red Star Award. See Table 1 for the specific distribution.

**Table 1.** Primary and secondary indicators for the evaluation system on the influence of the design industry on the carbon emissions of the manufacturing industry.

| | Primary Indicators | Secondary Indicators |
|---|---|---|
| The first- and secondary-level indexes of the evaluation system of the influence of design industry on carbon emission of manufacturing industry | Design resource size index | Number of design enterprises (enterprises) |
| | | Number of full-time design practitioners (persons) |
| | Economic efficiency index | Number of design achievement transactions (PCS) |
| | | Output value of design achievement transformation (ten thousand RMB) |
| | Government support capacity | Provincial support funding (ten thousand RMB) |
| | | Local matching funding (ten thousand RMB) |
| | Innovation composite index | Number of patents granted (PCS) |
| | | Major awards at home and abroad (number) |

We can use Equation (1) to normalize the specific data of the collected second-level indicators and obtain the corresponding standardized data. The normalized data of each second-level indicator $X_{ij}$ is $Y_{ij}$. Then, according to Equation (3), we determine the sample information entropy $E_j$ corresponding to each second-level indicator. Here, $P_{ij}$ is calculated using Equation (2) according to the $Y_{ij}$ value of each parameter in the sample. Then, using

Equation (4), we calculate the corresponding weight $W_i$ for each of the secondary-level indicators based on the information entropy $E_i$ of the second-level indicator.

$$Y_{ij} = \frac{X_{ij} - minX_i}{maxX_i - minX_i} \tag{1}$$

$$P_{ij} = \frac{Y_{ij}}{\sum_{i=1}^{n} \sum_{j=1}^{m} Y_{ij}}, \ (i = 1, \ldots, n; j = 1, \ldots m) \tag{2}$$

$$E_j = -\frac{1}{\ln(n)} \sum_{i=1}^{n} P_{ij} \ln P_{ij} \tag{3}$$

$$W_i = \frac{1 - E_i}{k - \sum E_i} \tag{4}$$

### 2.2. Ridge Regression Model

Although the entropy method can be used to determine the weight of the impact of each level of indicators, that does not mean the design industry necessarily has a significant impact on manufacturing carbon emissions. However, we believe that further research on the influence of the former on the latter is of sufficient research value since the design stage influences most of the changes in the manufacturing process. Therefore, further analysis is required to determine whether the design industry does influence changes in manufacturing carbon emissions. Furthermore, the relationship between positive and negative variables cannot be directly calculated through the entropy method model [52]. Thus, in this paper, we established a ridge regression model to judge whether the development of the design industry can explain the causes of carbon emission changes in the manufacturing industry.

Based on the original linear regression in Equation (5), $Y$ is the observation vector of the dependent variable with a dimension of $n \times 1$, $X$ represents the observation matrix of the independent variable, and its dimension is $n \times (p + 1)$. Also, $\beta$ signifies the vector coefficient at $p + 1$, while $\varepsilon$ denotes the random vector of $n$ dimension. We obtain the ridge regression model by adding a set of normal numbers to the diagonal of matrix $X'X$ (Equation (6)). Here, $I_{p+1}$ represents the identity matrix and $k$ is the ridge parameter. We can obtain a stable $k$ value according to the change trajectory of its selected value, and then substitute it as a fixed constant for regression analysis to obtain the corresponding calculation results.

$$Y = X\beta + \varepsilon, \tag{5}$$

$$\hat{\beta}_{RR} = \left(X'X + kI_{p+1}\right)^{-1}X'Y. \tag{6}$$

### 2.3. Structural Equation Model (SEM)

Since the regression model does not effectively reflect causes and effects, we cannot determine the causal relationship between the relevant indicators in the design and manufacturing industries through a ridge regression analysis alone. Therefore, to better study the relationship between carbon emissions in design and manufacturing and provide theoretical support on relevant policies, we propose a structural equation model. Besides, based on the normalization of each index in Equation (1), we further analyzed the relationship between them.

According to the relationship between the current indicators to be calculated and the carbon emissions of the manufacturing industry, we constructed a preliminary model relationship diagram (Figure 1).

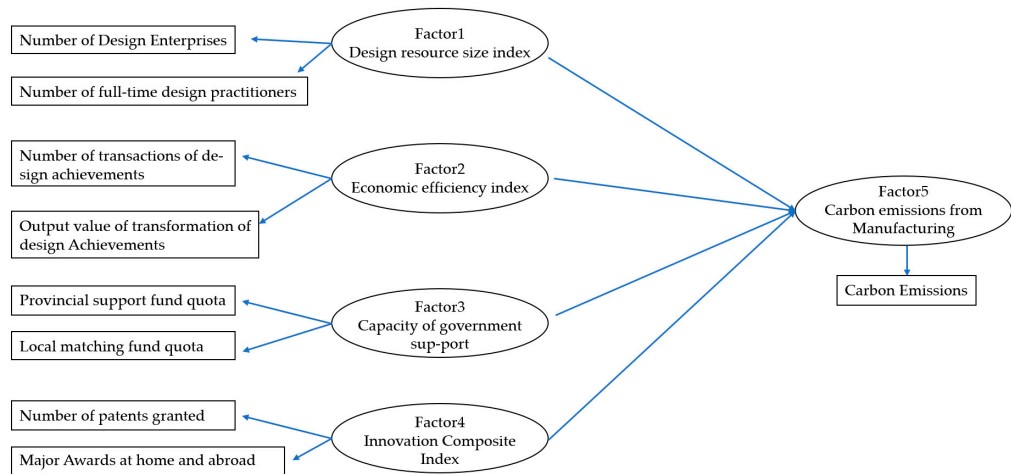

**Figure 1.** Diagram of the model illustrating the relationship between the industrial indicators and manufacturing carbon emissions.

### 2.4. Data Sources

The data sources in this paper are based on the 2016–2020 China Industrial Statistical Yearbook, China Energy Statistical Yearbook, Zhejiang Provincial Government Design Industry Dynamic Detection Summary, the Zhejiang Province Characteristic Design Demonstration Base Construction Monthly Summary, and the Zhejiang Industrial Production Data Summary.

## 3. Results

### 3.1. Weight Calculation Results

According to the above equation, we selected 18 cities in Zhejiang province where the design industry is relatively developed, namely Hangzhou, Ningbo, Wenzhou, Huzhou, Haining, Shaoxing, Yongkang, Quzhou, Zhoushan, Taizhou, Lishui, Yiwu, Tongxiang, Xiaoshan, Yuhang, Zhuji, Yueqing, and Jiangshan. We extracted relevant data of these cities from 2016–2020 for the analysis. In this study, we established four specific first-level indicators, and after performing weight calculations we used the entropy method and the regression model to verify the reliability of the weights obtained. Subsequently, we subdivided these factors into eight specific second-level indicators to monitor the corresponding data. According to Equation (1), after normalization we substituted the data of all second-level indicators into Equations (2)–(4) to accurately calculate the weights of all of the second-level indicators. Then, we added the corresponding second-level indicators together to obtain the weights of the first-level indicators. The weights of all of the indicators from 2016 to 2020 obtained using the entropy method were summarized to obtain the weight distribution map of secondary indicators displayed in Figure 2. Table 2 provides detailed figures that correspond to the distribution map in Figure 2.

Results indicate that among the four first-level indicators, the design resource scale index has the least impact on carbon emissions in the manufacturing industry, with a weight of only 7.6%. In contrast, the economic benefit index has the largest weight among the first-level indicators (32.85%) and thus the greatest impact on carbon emissions. Besides, the weight of government support is 29.67%, which is only marginally less important than the innovation composite index (30.14%). Among the secondary indexes, the index weights of the two design resource factors (industrial design companies and workers engaged as designers) both display a downward trend. This reflects the declining size and importance of the design industry itself, and its influence on carbon emissions continues to fall. Additionally, the number of patent licenses is the only other secondary index that exhibits a continuous declining trend, while the remaining indexes experienced yearly fluctuations. Therefore, we need to further analyze the weights obtained by the regression model analysis and combine them with corresponding annual manufacturing carbon

emissions data to confirm the weight of each index based on a stability analysis of the primary indicator weights. By using the entropy method mentioned above, and taking into account the data available in the current Energy Statistical Yearbook, we comprehensively calculated the weights of the four first-level indicators from 2013 to 2020, as Table 3 and Figure 3 show.

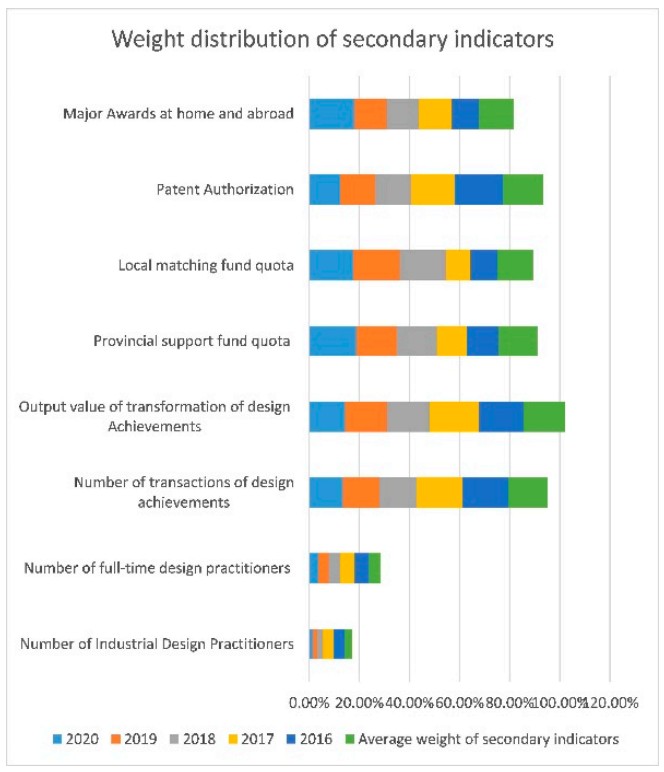

**Figure 2.** Weight distribution diagram of secondary indicators.

### 3.2. Ridge Regression Analysis

To test the rationality of the entropy method the weight calculations, we introduced the ridge regression model. Combined with the calculated carbon emissions of the manufacturing industry, the ridge regression model tests the reliability and the positive and inverse of each index weight. According to the carbon emissions of the manufacturing industry from 2013 to 2020, we adopted the common carbon emission formula from the IPCC method [53], which is expressed in Equation (7):

$$CO_{2e} = \sum_{i=1}^{n}(E_i \times NCV_i \times Cc) \times 10^{-3} \times COF \times \frac{44}{12}. \tag{7}$$

In Equation (7), $CO_{2e}$ signifies the carbon emissions of the manufacturing industry, $E_i$ represents the consumption of the $i$th energy, $NCV_i$ denotes the net calorific value of the $i$th energy, $Cc$ is the carbon content per unit calorific value, and $COF$ represents the oxidation factor, where the default is complete oxidation, i.e., $COF = 1$. Table 4 displays the results of the calculations and Figure 4 presents a graph to illustrate the results. We plotted the relevant data and added a polynomial asymptotic curve of order 2.

**Table 2.** Weight summary of all of the indicators from 2016 to 2020.

| Level-1 Indicator | Level-2 Indicator | 2020 Weight | 2019 Weight | 2018 Weight | 2017 Weight | 2016 Weight | Average Value of Level-2 Indicator | Average Value of Level-1 Indicator |
|---|---|---|---|---|---|---|---|---|
| Design resource size index | Number of design Enterprises (enterprises) | 1.59% | 1.75% | 2.04% | 4.29% | 4.50% | 2.83% | 7.60% |
| | Number of full-time design practitioners (persons) | 3.85% | 3.93% | 4.67% | 5.65% | 5.76% | 4.77% | |
| Economic efficiency index | Number of transactions of design achievements (PCS) | 13.40% | 14.60% | 14.90% | 18.25% | 18.30% | 15.89% | 32.85% |
| | Output value of design achievement transformation (ten thousand yuan) | 14.34% | 16.82% | 16.82% | 19.70% | 17.81% | 16.69% | |
| Government support capacity | Provincial support funding (ten thousand yuan) | 18.82% | 16.05% | 16.05% | 11.91% | 12.66% | 15.56% | 29.67% |
| | Local matching funding (ten thousand yuan) | 17.66% | 18.47% | 18.47% | 9.76% | 10.82% | 14.11% | |
| Innovation composite index | Number of patents granted (PCS) | 12.28% | 14.19% | 14.19% | 17.31% | 19.52% | 15.86% | 30.14% |
| | Major awards at home and abroad (number) | 18.06% | 12.87% | 12.87% | 13.12% | 10.63% | 14.28% | |

**Table 3.** Weight of the indicators at each level from 2013 to 2020.

| Year | Design Resource Size Index | Economic Efficiency Index | Government Support Capacity | Innovation Composite Index |
|---|---|---|---|---|
| 2013 | 11.03% | 35.21% | 19.97% | 33.79% |
| 2014 | 10.67% | 34.75% | 22.19% | 32.39% |
| 2015 | 10.44% | 35.29% | 20.45% | 34.15% |
| 2016 | 10.26% | 36.11% | 23.48% | 30.15% |
| 2017 | 9.94% | 37.95% | 21.67% | 30.43% |
| 2018 | 6.71% | 31.72% | 34.52% | 27.06% |
| 2019 | 5.68% | 29.40% | 32.22% | 32.70% |
| 2020 | 7.60% | 32.85% | 29.41% | 30.14% |

We chose carbon emission data as the dependent variable and the weight of the corresponding first-level indexes over the selected period as the independent variable for ridge regression analysis. Figure 5 shows the corresponding ridge trace map.

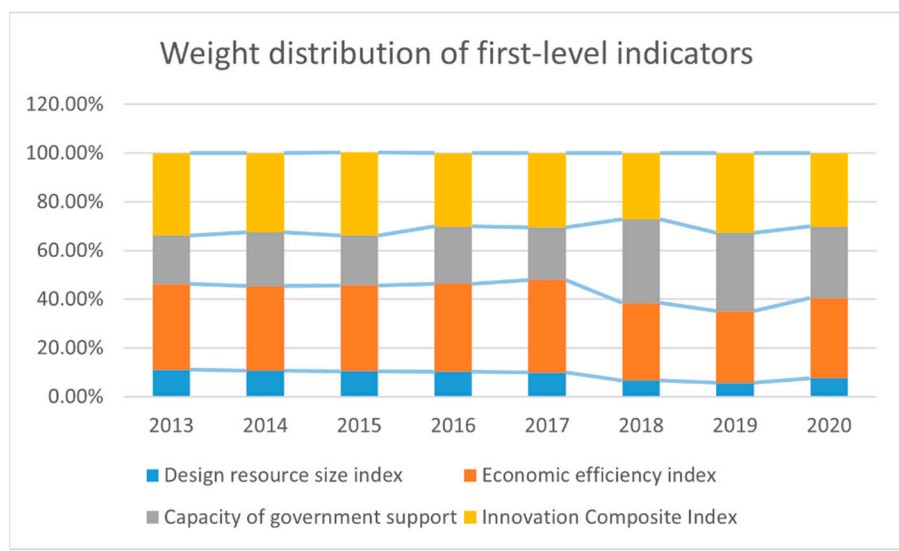

**Figure 3.** Weight distribution of the first-level indicators.

**Table 4.** Carbon emission data of Zhejiang Province, 2013–2020.

| Year | Carbon Emissions (Ten Thousand Tons) |
| --- | --- |
| 2013 | 37,280 |
| 2014 | 37,652 |
| 2015 | 39,220 |
| 2016 | 40,552 |
| 2017 | 42,060 |
| 2018 | 43,350 |
| 2019 | 44,786 |
| 2020 | 46,339 |

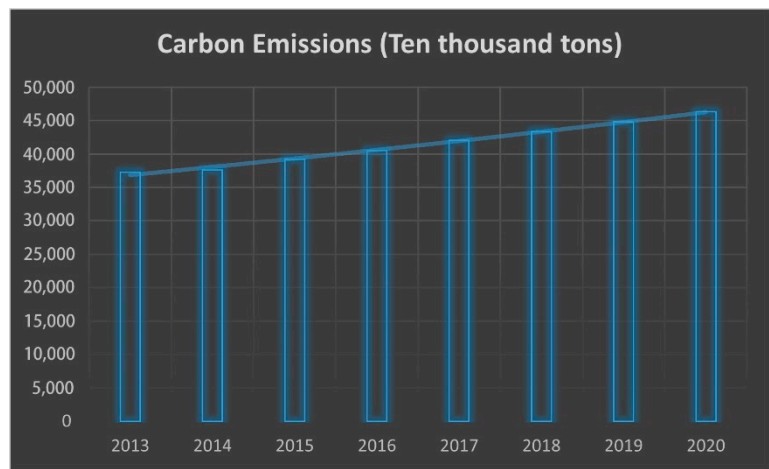

**Figure 4.** Carbon footprint map.

According to the ridge regression diagram, when the k value is 0.01, the normalized regression coefficients of each variable are stable, so the optimal value for k is 0.01. Table 5 presents the results after the implementation of the ridge regression model.

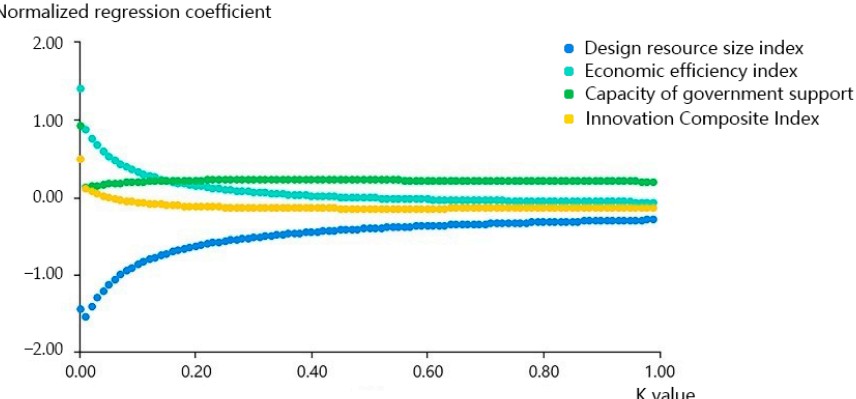

**Figure 5.** Ridge trace figure.

**Table 5.** Ridge regression analysis results.

| | Unstandardized Coefficients | | Standardized Coefficients | t | p | $R^2$ | Adjusted $R^2$ | F |
|---|---|---|---|---|---|---|---|---|
| | B | Standard Error | Beta | | | | | |
| Constant | 24,554.392 | 8844.280 | - | 2.776 | 0.109 | | | |
| Design resource size index | 20,822.462 | 35,909.676 | −1.560 | −5.799 | 0.028 | | | |
| Economic efficiency index | 86,491.702 | 22,260.913 | 0.871 | 3.885 | 0.036 | 0.978 | 0.933 | F(4,2) = 21.834 p = 0.044 |
| Capacity of government support | 6272.282 | 6555.546 | 0.130 | 0.957 | 0.440 | | | |
| Innovation composite index | 13,090.373 | 15,030.795 | 0.114 | 0.871 | 0.476 | | | |

According to the results in Table 5, the F-test value of $p = 0.044$ is less than 0.05, which indicates that the model is meaningful. After obtaining these comprehensive results, we can determine the following relationship: manufacturing carbon emissions = 24,554.392 − 208,222.462 × design resource scale index + 86,491.702 × economic benefit index + 6272.282 × government support capacity + 13,090.373 × innovation composite index.

Table 6 presents a summary of the results, according to the existing ridge regression model.

**Table 6.** Ridge regression model summary.

| Model Summary | | | |
|---|---|---|---|
| Sample Size | $R^2$ | Adjusted $R^2$ | Model Error |
| 7 | 0.9776 | 0.933 | 737.722 |

The R square value in the table is 0.978, which means that the scale index of design resources, economic benefit index, government support capacity, and innovation composite index can explain 97.76% of the changes in carbon emissions of the manufacturing industry, from the perspective of the design industry. Thus, the design stage of the manufacturing process has a tremendous impact on its carbon emissions. Therefore, it is of practical value to further study the impact of design industry development on carbon emissions in manufacturing.

### 3.3. Structural Equation Model Analysis

We selected the first-level indicators and the calculated carbon emissions of the manufacturing industry in Zhejiang province as potential variables, and the second-level indicators associated with the first-level indicators as the corresponding measurements of the potential variables. Relevant data were substituted into the calculations to obtain preliminary results. First, we analyzed the obtained model fitting indicators by applying six commonly used fitting judgment indicators. The results are displayed in Table 7.

**Table 7.** Structural equation model preliminary fitting index results.

| Commonly Used Indicators | GFI | RMSEA | RMR | CFI | NFI | NNFI |
|---|---|---|---|---|---|---|
| Requirement | > 0.9 | < 0.10 | < 0.05 | > 0.9 | > 0.9 | > 0.9 |
| Calculated result | 0.996 | 0.247 | 1.034 | 1.012 | 0.883 | 0.950 |

Table 7 indicates that the RMSEA value exceeds the requirement of the judgment criterion of <0.1, the RMR value also exceeds the criterion of <0.05, and the NFI value does not meet the requirement of >0.9. Overall, three of the six most commonly used judgment indexes of model fitting level are unsatisfactory, indicating that the model fitting combination is not ideal. As a result, we made further adjustments to the coefficients and relationships to refit the model. Based on the results of the calculations, Table 8 presents the relationship and MI value distribution among all of the paths in the model.

**Table 8.** Path impact relationship distribution.

| X | (X Impacts Y) | Y | MI |
|---|---|---|---|
| Factor 4 | - | Factor 5 | −0.392 |
| Factor 2 | - | Factor 5 | 0.370 |
| Factor 3 | - | Factor 5 | 3.487 |
| Factor 4 | - | Factor 1 | 0.865 |
| Factor 3 | - | Factor 2 | 12.347 |
| Factor 2 | - | Factor 1 | 7.528 |

Table 8 indicates that Factor 3 has a significant influence on Factor 2, while Factor 2 influences Factor 1. Besides, the MI index value of the Factor 3 influence on Factor 2 is larger than 10. Thus, we can theoretically enhance the model further. Therefore, Factor 3 affects Factor 2 and Factor 2 influences Factor 1 in the relationship. We increased the value of MI to be greater than 10 to improve the model, and Table 9 presents the results of the fitting indexes obtained by the model after subsequent adjustments.

**Table 9.** Model fitting analysis results.

| Commonly Used Indicators | GFI | RMSEA | RMR | CFI | NFI | NNFI |
|---|---|---|---|---|---|---|
| Requirement | > 0.9 | < 0.10 | < 0.05 | > 0.9 | > 0.9 | > 0.9 |
| Calculation results | 1.042 | 0.035 | 0.043 | 1.327 | 0.952 | 1.030 |

In Table 9, we can see that the model fitting results have satisfied all of the judgment indexes, so we can conclude that the model fitting degree is relatively good. Thus, we obtained revised results for the regression coefficients of the model, as Table 10 and Figure 6 illustrate below.

Based on the distribution of standardized regression coefficients in Table 10 and Figure 6, we can conclude that the scale index of design resources has a restrictive effect on carbon emissions in the manufacturing industry. Conversely, the economic efficiency index, government support capacity, and innovation composite index all have a positive influence on the carbon emissions of the manufacturing industry According to Figure 6, in

the weight distribution of indicators affecting carbon emissions of manufacturing industry, the economic benefit index is the largest, followed by the innovation composite index, government support capacity, and design resource scale index. These results correspond to the weight distributions calculated by the entropy method.

**Table 10.** Regression coefficient summary.

| X | Y | Non-Standardized Regression Coefficient | SE | z | p | Standardized Regression Coefficient |
|---|---|---|---|---|---|---|
| Factor 1 | Factor 5 | −6.527 | 1.416 | −1.478 | 0.023 | −0.135 |
| Factor 2 | Factor 5 | 7.650 | 2.009 | 2.798 | 0.039 | 0.994 |
| Factor 3 | Factor 5 | 0.330 | 0.384 | 0.860 | 0.390 | 0.225 |
| Factor 4 | Factor 5 | 16.459 | 0.756 | 7.073 | 0.409 | 0.975 |
| Factor 3 | Factor 2 | 0.582 | 0.678 | 0.858 | 0.391 | 0.821 |
| Factor 2 | Factor 1 | 2.198 | 2.178 | 1.009 | 0.313 | 0.999 |
| Factor 1 | Number of design enterprises | 1.000 | - | - | - | 0.901 |
| Factor 1 | Number of full-time design practitioners | −0.958 | 0.267 | −3.584 | 0.007 | −0.963 |
| Factor 2 | Output value of design achievement transformation | −2.257 | 2.171 | −1.039 | 0.299 | −0.992 |
| Factor 2 | Number of design achievement transactions | 1.000 | - | - | - | 0.462 |
| Factor 3 | Provincial support funding | 1.000 | - | - | - | 0.666 |
| Factor 3 | Local matching funding | 1.494 | 0.850 | 1.757 | 0.079 | 0.982 |
| Factor 4 | Number of patents granted | 1.000 | - | - | - | 0.010 |
| Factor 4 | Major awards at home and abroad | 9.388 | 16.778 | 5.924 | 0.000 | 0.996 |

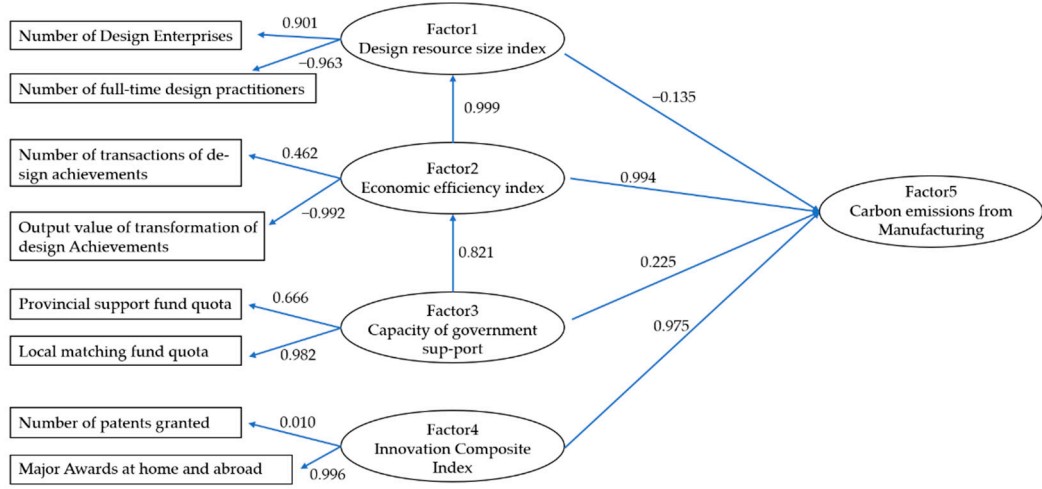

**Figure 6.** Diagram of model results.

As Figure 6 indicates, there is also a significant relationship among the various indicators. For the indicators within each level, an increase in government support has a positive effect on the economic benefit index, while growth in the economic benefit index results in the enhancement of the design resource scale. Between the primary and secondary indicators, a larger number of design enterprises leads to an increase in the design resource scale index, while a greater number of design professionals somewhat inhibits

development of the design resource scale. In terms of economic benefits, an increase in the number of design transactions leads to a slight improvement in economic benefits, but development of the output value of transformational design does not lead to growth in economic benefits. Concerning government support, funding from local and provincial governments has a significant role in stimulating the development of the design industry. Regarding the innovation composite index, higher numbers of patents granted and awards given promote the innovative ability of the design industry, to a certain extent. Besides, the number of major domestic and international awards correlates with improvements in industrial innovation ability.

In terms of the distribution of each index, the economic benefit index and the innovation composite index have the greatest impact on the development of the design industry and the carbon emissions of the manufacturing industry. Their coefficients are much greater than the other two first-level indexes, which confirms that the design industry is a progressive sector that is highly dependent on innovation. Comparatively speaking, expansion of industrial scale has a restrictive effect on carbon emissions in the manufacturing industry, indicating that the design process itself promotes the transformation and advancement of the manufacturing industry. Additionally, appropriate design planning during the initial stages of the production chain has positive significance for carbon emission reduction in the manufacturing phase. For each level of indicators, the importance of government support is reflected in the impact on the economic benefits of the design industry, rather than the impact on the carbon emissions of the manufacturing industry. Government support for the design industry greatly enhances economic benefits, thus further promoting the expansion of design industry scale and related resources, and stimulating industry development.

## 4. Conclusions

### 4.1. Discussion of Research Results

The design industry belongs to the high-tech creative sector and relies heavily on scientific and technological innovation. Its survival and development greatly depend on the transformation of the economic benefits of innovation content. Moreover, the design industry obtains advanced industrial capital through the efficient economic transformation of cultural creativity, emerging technologies, and other sources [54–56]. With the rapid development of the new internet era, this type of industry creates high added value by connecting with other corresponding industries, becoming a main driving force that develops national strategic brands and technological innovation [57]. The design industry itself is closely linked to the manufacturing industry. Through reasonable design, the added value of products can be enhanced to increase sales and achieve higher commercial value. The manufacturing process involves heavy industry and its carbon emissions are generally high, and the product decisions made in the design stage have a direct impact on carbon emissions during the production process [58]. After the design stage of a certain product, there are several choices regarding production procedures. Since the corresponding carbon emissions of each procedure are different, the design stage naturally has a significant impact on emissions [59].

In this paper, we deconstructed several factors of the design industry that affect the carbon emissions of the manufacturing industry. We devised corresponding primary and secondary indexes and calculated the weight of each relevant index using the entropy method. We also introduced a ridge regression model to aid our analysis. Based on the results obtained, we established that design industry development can effectively explain most of the reasons for changes in manufacturing carbon emissions. More specifically, 97.79% of the changes in manufacturing carbon emissions can be explained from the perspective of design industry development. Therefore, our study on the relationship between the two industries has great practical value. In this study, we re-introduced the structural equation model and used as factors the first level indexes and carbon emissions of the manufacturing industry. We obtained a revised structural equation model by adding an influence path and appropriately amplifying the MI value. Modeling results revealed

that expansion of the scale index of design resources has a restrictive effect on the carbon emissions of the manufacturing industry. The economic benefit index, government support capacity, and innovation composite index of the design industry all promote carbon emissions in the manufacturing industry. The weight distribution obtained by the structural equation is consistent with the entropy method, meaning that the weight of the economic benefit index is greater than the innovation composite index, while government support capacity outweighs the design resource scale index.

The first-level index planned in this paper is closely related to the development of the design industry itself. The economic benefit index has the largest weight since higher indexes result in greater market satisfaction toward designed products, which reflects the impact this index has on carbon emissions. As a result, the manufacturing industry produces a larger number of products. This expansion of the production base subsequently leads to an increase in carbon dioxide emissions [60]. Dominated by the market profit mechanism, the design industry tends to be solely market-focused. It aims to gain higher profits, but often ignores the issue of environmental protection [61]. This consequently leads to a rise in carbon emissions during the production process. As for green design, there are no appropriate laws or regulations in China to encourage environmental protection in the design stage, and designers are often unaware of green design. Thus the carbon emissions of the manufacturing industry increase with the rising economic benefit index [62]. The influence of government support and the innovation composite index on carbon emissions is positive but insignificant, and the weight of the two is less than the economic benefit index. The reason for this is that both of them promote the development of the design industry. In related studies, the development of the design industry is extremely dependent on its innovative capability [63], and several studies have shown that government support is one of the core driving forces for its development [64,65]. These two indicators promote the development of the design industry and expand the scale of production in the corresponding manufacturing industry. Currently, green design has not penetrated the ethical professional system of designers [66–68], the vast scale of production will generate huge carbon emissions. Design resource scale index in the calculation of entropy value method, its minimum weight significantly, in fact reflects that the design industry demand for design direction of resource is not unbearable, and the number of design enterprise is not decisive in the manufacturing industry regarding carbon emissions as the influence of the design of carbon emissions is reflected in the design of products and market to cater to the degree of quality and design itself. No amount of companies can scale up their manufacturing operations without designing excellent products. And its inhibition, due to the expansion of design resource, will let now advocate green design direction of resources into the practice of design, and it will also attract more education practitioners with stronger green design consciousness into the system design, and thus more production process with less carbon emissions, which bodes well for natural inhibition of carbon emissions. The above weight distribution of the influencing factors of the design industry on the carbon emissions of the manufacturing industry is in essence consistent with the industrial characteristics of the design industry itself, which is the reality that high-tech and creative industries attach importance to creativity rather than limited to the industrial scale.

### 4.2. Research Conclusions

Given the results of our research, we can provide practical suggestions for government decision-making related to the design industry. First, carbon emission reduction in the manufacturing industry from the perspective of carbon neutrality is critical. Moreover, the design industry has an immense impact on the carbon emissions of the manufacturing industry, which is a point that cannot be ignored when devising carbon emission reduction measures. In the design industry, the orientation of design resources should be regulated, and more human, cultural, and social resources should be directed toward the correspond-

ing industry. This makes full use of the restraining effect of the scale of design resources on carbon emissions in manufacturing.

According to regulations and the control of industrial development, and regarding both its economic benefit index and creative composite index, the goal of the government is to promote the development of industry. At the same time, it should avoid unchecked development, which leads to sharp increases in manufacturing carbon emissions. The authorities should vigorously support the development of advanced science and technology creative industries and strengthen the related design industry based on corporate social responsibility. The concept of green design will then further penetrate industrial development.

In the process of design education, it is necessary to cultivate the sense of designers' social and moral responsibility, integrate the concept of green design into the education system, and vigorously cultivate "green design talents". We should expand the scale of design resources, thereby restricting carbon emissions. Additionally, the participation of these talents will reduce the influence of government support, economic efficiency index, creative composite index, and other indicators to cut carbon emissions.

Furthermore, it is necessary to be cautious when adjusting the economic benefit index in the design industry. Among the factors affecting carbon emissions of the manufacturing industry, this item carries the largest weight, but it is also directly related to the dynamics of future industrial development. Therefore, the development of high-tech creative industries should be supported and the economic benefits improved. In practical applications, the economic benefits of the design industry should involve more evaluation norms, such as corresponding evaluations of the green index of design results, the establishment of an evaluation system of the green index of the design, etc., to promote a reduction in carbon emissions and the healthy development of the industry.

**Author Contributions:** Conceptualization: B.X. and H.Q.; methodology: B.X. and H.Q.; formal analysis: B.X. and H.Q.; writing—original draft preparation: B.X. and H.Q.; writing—review and editing: B.X. and H.Q.; project administration: B.X. and H.Q.; funding acquisition: B.X. and H.Q. All authors have read and agreed to the published version of the manuscript.

**Funding:** This research was financially supported by the Collaborative Education Project of Academia-Industry Cooperation of the Ministry of Education of China (No. 202102233002).

**Institutional Review Board Statement:** Not applicable.

**Informed Consent Statement:** Not applicable.

**Data Availability Statement:** Data are available from the corresponding authors upon reasonable request.

**Conflicts of Interest:** The authors declare no conflict of interest.

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
