# Peer review of "Impact of the Design Industry on Carbon Emissions in the Manufacturing Industry in China: A Case Study of Zhejiang Province"

_sustainability, doi:10.3390/su14074261_

Round 1

Reviewer 1 Report

I do not understand the aim of this paper and I do not understand why authors named they formula extended Kaya identity. If they want to assess carbon emission reduction in manufacturing industry they can apply productivity with undesirable outcomes (GHG emissions) approach in manufacturing industry. Kaya identity was designed more for global  economy or country level data, to show simple  macro-level drivers of GHG emissions: GDP per capita, energy intensity per unit GDP and the emissions per unit energy. The authors should explain better why hey decided to call they approach as extended Kaya identity in manufacturing industry. In addition, literature review on Kaya identity is very weak and important references are missing. I am very skeptical about this paper, hope authors can revise it and provide answers. 

Author Response

请参阅附件

Reviewer 2 Report

The title could be shortened in order to focus on the main purpose.

The abstract shoud be reviewed in order to place the question addressed in a broad context and highlight the purpose of the study. Apart from summarizing the article’s main findings, the main conclusions and interpretations should be indicated.

In the introduction, Chinese specific goals and objectives are not included. At the same time, the gap to be covered/solved is not stressed. Nor any nolvety is highlighted. This must be deeply improved.

The structure of the literature review must be enhanced.  Which is the gap in the literature that this paper tries to solver/cover?

The purpose of the equations introduced in the methods should be stressed. Which is the relation between these equations and the results provided? Which is your interpretation? If authors want to stablish a relation, linear regressions should be avoided. Causality should be added. SEM analysis could be included. In addition, goodness and reliability of the model should be proved.

Conclusions should be reviewed. First, stablishing the discussion of the results. Second, stating the main conclusions of the research.

Reviewer 3 Report

Dear Authors,

Thank you for the opportunity to read your scientific article.

This is an interesting and important article focused on the current important topic of the impact of design end factors on carbon emission reduction in the manufacturing industry.

From a methodological point of view, it suitably combines regression and correlation analysis, Logarithmic Meam Division Index and Shann information entropy analogy.

I have only two remarks:    

Formula (6) p. 6 in the fraction's denominator, the variable Y is indexed ij, so a double sum should be formally used.
Formula (7) page 6 - why is the Shannon entropy constant equal to ln (1 /n)?

Round 2

Reviewer 1 Report

The authors did good job and corrected heir manuscript. My comments were addressed in revised version of manuscript and answers were provided. 

Author Response

请参阅附件。

Reviewer 2 Report

P1. The title has been improved. This is interesting.

P2. The abstract has been improved.

P3. The introduction has been improved. The gap to be covered is well raised.

P4. The answer is adequate. The intemption is good. However, the literature review is too short to be an independet section. I suggest to enlarge it or join with the introduction section. It can be noted that this second alternative is the one proposed by the journal in its recommendations.

P5. Strategy is adequate. However, in page 7 there is a empty table. In addition, Figure 7 should be normalized. Values between -1/0/1 should be provided. Which is the purpose of the equation 8 at the end of results? More development is needed at the end of this section.

P6: Ok.

Round 3

Reviewer 2 Report

P4. Ok.

P5. Ok.

The paper has been improved.

Congrats!